# The Role of Fetuin-A in Tumor Cell Growth, Prognosis, and Dissemination

**DOI:** 10.3390/ijms252312918

**Published:** 2024-11-30

**Authors:** Peace Odiase, Jonathan Ma, Sruthi Ranganathan, Olugbemiga Ogunkua, Winston B. Turner, Dana Marshall, Josiah Ochieng

**Affiliations:** 1Department of Biochemistry, Cancer Biology, Neuroscience and Pharmacology, Meharry Medical College, Nashville, TN 37208, USA; opeace20@email.mmc.edu (P.O.); oogunkua@mmc.edu (O.O.); wturner24@mmc.edu (W.B.T.); 2College of Arts and Science, Vanderbilt University, Nashville, TN 37203, USA; jonathan.j.ma@vanderbilt.edu; 3Department of Medicine, Cambridge University, Cambridge CB2 OQQ, UK; sr932@cam.ac.uk; 4Department of Pathology, Meharry Medical College, Nashville, TN 37208, USA; dmarshall@mmc.edu; 5Department of Biomedical Science, School of Graduate Studies Meharry Medical College, Nashville, TN 37208, USA

**Keywords:** Fetuin-A, 3-D growth, metastasis, liver, breast, prostate, lung, cancer

## Abstract

Fetuin-A, also known as alpha-2-Heremans-Schmid-glycoprotein (Ahsg), is a multifunctional molecule with diverse roles in biological processes such as mineralization, tumor growth, and inflammation. This review explores the involvement of Ahsg in various cancers, including liver, breast, prostate, colorectal, brain, osteosarcoma, and lung cancers. In many cancer types, Ahsg promotes tumor growth, invasion, and metastasis through various mechanisms, including cellular adhesion, spreading, chemotaxis, and modulation of cell-growth signaling pathways. Additionally, Ahsg has been implicated in the regulation of inflammatory cytokine production, making it a potential marker of inflammation in cancer. The complex interplay between Ahsg and cancer progression highlights its potential as a diagnostic biomarker and therapeutic target in various cancers. However, further research is needed to fully elucidate the mechanisms of action of Ahsg in cancer and to explore its clinical implications in cancer diagnosis, prognosis, and treatment.

## 1. Introduction

Fetuin-A, also known as alpha-2-Heremans-Schmid (HS)-glycoprotein (Ahsg), is a 64 kDa plasma protein that is part of the cystatin family of cysteine protease inhibitors [1]. Ahsg is involved in numerous biological functions including the regulation of calcification, tumor growth, endothelial cell function, calcium metabolism, senescence, protein metabolism, insulin signaling, and angiogenesis [1,2,3,4,5,6]. Classically recognized as a hepatokine, Ahsg regulates insulin function by suppressing the auto-phosphorylation of insulin receptor tyrosine kinase [3,7]. Due to its proinflammatory attributes, Ahsg has been shown to stimulate the production of inflammatory cytokines in adipocytes and macrophages, making it a marker of inflammation [8,9]. However, Ahsg can also function as an anti-inflammatory marker in various diseases, such as endotoxemia, sepsis, pancreatitis, chronic kidney disease, and rheumatoid arthritis [10,11,12]. Its opsonizing properties aid in phagocytosis and pathogen clearance, contributing to innate immunity [13]. Ahsg is synthesized by the liver and secreted into the blood where its concentration is relatively high (~0.4–0.6 mg/mL) [14]. Interestingly, Ahsg accumulates in the bone matrix over time, where its concentration can be higher than that in the blood [1]. While Ahsg expression is typically restricted to the liver, its expression can be induced during tumor progression [15,16,17,18]. Initially, this ectopic synthesis of Ahsg did not make sense given that there is enough protein in the blood. However, more recent studies have solidified its role as a tumor antigen in several cancer types. Thus, autoantibodies targeting ectopically synthesized Ahsg are valuable as prognostic biomarkers [19]. Most, if not all, of the processes modulated by Ahsg in tumor growth and dissemination can be attributed to the protein interacting partners of Ahsg (Figure 1A). For example, Ahsg is an established ligand for TLR4 [20]. It also interacts with histones, particularly histone H2A, that most likely interact with the D-domain of Ahsg, which is rich in negatively charged amino acids [1,21,22].

## 2. Cell Growth

Ahsg is implicated in pathways that directly or indirectly promote cell growth and involve numerous mechanisms, including (i) activation of the phosphatidylinositol 3 (PI3) kinase/Akt signaling pathway and (ii) mimicry of the Transforming growth factor-β (TGF-β) receptor, enabling involvement in growth signaling pathways [23]. First, Ahsg can activate the PI3k-Akt pathway, which is involved in cellular growth and proliferation [24]. Downstream targets of this pathway include mTOR, MDM2, GSK3, and FOXO, which can all promote cell growth, survival, and metabolism by influencing gene expression [25]. For instance, prevention of FOXO entry into the nucleus upon activation of the PI3k-Akt pathway prevents the expression of AR, ERG, and Runx2, which then promotes cell growth and cancer [26]. The activation of the PI3k-Akt pathway by Ahsg requires annexins, particularly annexin-A2 and annexin-A6 [23]. Secondly, mimicry of the TGF-β receptor by Ahsg can enable modulation of a broad range of growth signaling pathways. While Ahsg can promote some growth-factor-mediated signaling pathways, it can reduce some others. One study showed that the downregulation of Ahsg enabled the promotion of TGF-β-receptor-mediated signaling and tumor growth [27]. Overall, Ahsg is able to modulate a wide range of signaling pathways by interacting with various molecules (Figure 1B) that can cause the increased expression of growth-promoting genes.

## 3. Invasion and Metastasis

The multifunctional role of Ahsg in cancer progression has emerged as a pivotal player in cancer invasion, potentially serving as a major player in tumor metastasis and a vital component in the formation of metastatic niches for tumor cells. Firstly, Ahsg is implicated in cancer invasion through its role in exosome mechanisms. Exosomes, which are a type of extracellular vesicle, play significant roles in the cellular mechanisms for growth and motility and the invasive capacity of tumor cells. They shuttle biomolecules such as miRNA and tRNA between cells to affect the normal physiology and pathophysiology of the recipient cells. On an intracellular level, Ahsg shuttles histones from the cell nucleus to exosomes, and the uptake of histone-coated exosomes by cells has been shown to drive focal adhesion assembly, cellular adhesion, and spreading on various extracellular matrices [28]. In serum, Ahsg was shown to robustly support directed chemoattraction and invasion by breast carcinoma cells through Matrigel-coated inserts. This effect of Ahsg also seems to synergize with the chemotactic abilities of stromal derived factor-1 (SDF-1) to mediate motility, providing evidence for the involvement of cellular exosome uptake mechanisms, since this synergy promoted adhesion via exosomes [21]. Furthermore, another study on breast carcinoma cells as well as colorectal cancer cells found that in the presence of Ahsg, the intracellular calcium ion concentrations in tumor cells increased, resulting in the secretion of exosomes containing Ahsg, plasminogen, and histones, which mediate adhesion and cell spreading. Proteomic analysis found that exosomes secreted in the absence of Ahsg lacked plasminogen and histones and thus lacked the capacity for mediating adhesion [22]. This interplay between histones and Ahsg was more specifically elucidated in a study that demonstrated that the breast carcinoma, glioblastoma, and prostate cancer cell uptake of exosomes was promoted upon incubation with Ahsg and histones but not with Ahsg alone. This uptake was mediated by surface syndecan-4 (SDC4)-mediated endocytosis and mediated motility and invasion [29]. Ahsg uptake likely has an important role in tumor cell invasion by increasing tumor cell adhesion to the extracellular matrix, thus facilitating their dissemination.

Secondly, the changes in the sialylation of Ahsg have been demonstrated to be potentially involved in promoting tumor motility and invasion. Glycoproteomic analysis of a doxorubicin-treated hepatocarcinoma cell line identified increased sialylation levels for Ahsg in cell lysates [30]. Additionally, sialic acid residues on Ahsg were found to be essential in promoting the progression of glioblastomas. This was demonstrated by showing that asialofetuin-A (ASF) acted as a dominant negative form of Ahsg, attenuating the uptake of Ahsg and modulating motility in glioblastoma cells [31]. Toll-like receptor-4 (TLR4) has been implicated in tumor progression, though the role of TLR4 in Ahsg uptake has only recently been explored [32]. Inhibition of the rapid endocytotic uptake of Ahsg was demonstrated in TLR4-knockdown prostate tumor cells treated with a TLR4-specific inhibitor (CLI-095), which inhibited tumor cell adhesion and invasion through Matrigel [33]. A third potential pathway through which Ahsg plays a role in invasion and metastasis involves the transforming growth factor-beta (TGF-β) signaling pathway. Specifically, Ahsg regulates the release of matrix metalloproteinase-9 (MMP-9), which could impact the degradation of the extracellular matrix [34]. Given the association between MMP-9 and cancer progression, the modulation of MMP-9 release by Ahsg may indirectly affect cancer invasion and metastasis [35]. Further research will be needed to explore these mechanisms and unveil other potential mechanisms regarding Ahsg’s role in cancer invasion and metastasis. Fourthly, Ahsg attracts tumor cells and promotes the biogenesis and secretion of bioactive exosomes, which in turn promote the 3D growth of tumor cells. Ahsg has been shown to have chemoattraction properties in in vitro chemoattraction assays [28,36] These properties may be in part responsible for some of the tumor homing processes depicted in Figure 2. For example, Ahsg in the blood vessels, including those in the tumor microenvironment, may facilitate the process of intravasation [37]. Since the concentration of Ahsg in the bone matrix is higher than in the blood vessels [2], this could also facilitate the process of the extravasation of tumor cells that home to the bone, such as prostate and breast (Figure 2). Similarly, Ahsg is concentrated in the liver (site of synthesis), relative to its concentration in the blood, and this likewise may facilitate the extravasation of tumor cells that preferentially home to the liver during the process of metastasis (Figure 2).

## 4. Synthesis of Ectopic Ahsg by Various Cancer Types

The serum levels of Ahsg are markedly elevated in patients with cancers (such as prostate cancer) compared to healthy individuals and/or benign controls [17,27,38,39,40,41,42,43,44,45,46,47,48,49,50,51,52,53]. The elevated serum levels most likely emanate from ectopic Ahsg synthesized by the growing tumor mass. This presents potential for clinical intervention, where Ahsg can serve as a diagnostic and prognostic biomarker for cancer. Therapeutic strategies targeting Ahsg or its downstream signaling pathways may further offer a means to inhibit tumor growth and metastasis.

### 4.1. Liver Cancer

Ahsg expression has been associated with liver cancer including hepatocellular cancer (HCC) and cholangiocarcinoma [54]. Serum concentrations of Ahsg are reported to be decreased in patients with hepatocellular cancer but not in other liver pathologies such as viral hepatitis from Epstein–Barr virus and acute alcoholic hepatitis [55]. However, it is noteworthy that decreased serum levels of Ahsg are found in patients with liver cirrhosis as well as those with HCC and therefore this is not specific to HCC. It is possible that changes in Ahsg are associated with chronic liver conditions such as cirrhosis and HCC, unlike acute conditions such as acute alcoholic hepatitis. This is further supported by other findings that show that the Ahsg levels in the serum were not correlated with acute inflammatory markers such as IL-6 or fever. The findings of this study are further reinforced by another study that reports the use of Ahsg as a marker of HCC risk, when patients with diabetes were excluded. Though the association of Ahsg in HCC was believed to involve metabolic pathways that are commonly dysregulated in diabetes, this novel association could be an indication of other supporting pathways for the involvement of fetuin A in HCC [54]. Moreover, it is interesting that Ahsg levels are found to be decreased in HCC since Ahsg is believed to activate the TLR4/JNK/NF-κB pathway, which is involved in inflammation, a response commonly associated with HCC [56]. In addition, though decreased Ahsg levels are associated with HCC and chronic liver pathologies, an increase in fucosylated Ahsg levels is also associated with HCC [42]. Fucosylated Ahsg is useful in other liver cancers such as cholangiocarcinoma. A study reports on the use of fucosylated Ahsg for differentiating cholangiocarcinoma from other liver pathologies, including primary sclerosing cholangitis; this method even performs well when compared with existing ‘gold standard’ cancer markers such as Ca 19-9 [42]. The authors showed that this modified Ahsg is potentially more predictive of cholangiocarcinoma than HCC and developed a diagnostic tool. Since liver cells make most of the Ahsg in the blood stream, it was interesting to observe that liver cancer cells (hepatocellular carcinoma) that produce the high levels of Ahsg (RNAseq) have a better overall survival compared to those that produce low levels of Ahsg, with a median survival of 71.03 months for high expression vs. 73 months for low expression [57,58].

### 4.2. Breast Cancer (Metastasizes Mainly to Bones, Liver, Lung, and Brain)

Ahsg has been linked to tumor growth, cellular adhesion, and chemotaxis in in vitro and in vivo studies of breast cancer cells. Mainly propelled by Ca^2+^ ions, in vitro, breast cancer cells adhered to immobilized Ahsg. This attachment was restricted to fractions containing Ahsg in fractionated human serum and was accompanied by PI3-kinase/Akt activation, which was reduced in cells lacking annexin-A6, a cell surface receptor for Ahsg [23]. The formation of stable complexes with histones in the cell nucleus and subsequent transport into exosomes by Ahsg plays a crucial role in the cellular adhesion and spreading and appears to be mediated by TLR4 [31,33]. TLR4 binds vesicle-associated histones, activating inflammation and macrophytic activity [59]. Furthermore, in the presence of CLI-095, a TLR4 inhibitor, the rapid endocytic uptake of Ahsg is inhibited, resulting in a decline in the adhesion of tumor cells as well as invasion through a bed of Matrigel [33]. Interestingly, the appearance of both Ahsg and TLR4 in the nucleus has been reported. The metastasis of breast cancer cells involves chemotaxis, which is highly driven by cellular adhesion. Ahsg plays a crucial role as a pivotal chemoattractant driving the invasion and trafficking of tumor cells through the extracellular matrix. Cell-to-extracellular matrix interactions, cellular adhesion, and spreading are impacted by the release of intracellular galectin-3, a β-galactoside binding protein that is regulated by Ahsg [60]. Moreover, Ahsg, in conjunction with the chemokine CXCL12 (also called stromal-derived factor-1 (SDF-1)), has been shown to enhance chemotaxis in in vitro studies of breast cancer cells [31].

In vivo studies reveal that Ahsg is a key modulator of mammary tumor progression and host survival. In TGFα-driven breast cancer incidence, reduced serum Ahsg contributed to a delay in breast cancer progression in FGFR4-deficient mice by altering metabolic pathways such as adipogenesis [61]. Likewise, in the polyoma middle T antigen (PyMT) transgenic mouse model, PyMT/Fet+/− mice showed significantly longer latency compared to the 90 days post-birth observed in the control group, PyMT/Fet+/+. Tumorigenesis was inhibited in PyMT/Fet−/− double-knockout mice for the total duration of the study (40 weeks) via upregulation of p19ARF, p53, and TGF-β1 signaling, while the PI3-kinase/Akt signaling pathway was unaffected [62]. Looking at overall survival in breast cancer, tumor cells that produce high levels of Ahsg, as measured by RNAseq, have a poorer survival compared to those that express low levels (Figure 3), with a median survival of 115 months for high expression vs. 148 months for low expression [58]. Taken together, the absence of Ahsg supported tumor suppression, cell cycle arrest, cellular senescence, and apoptotic pathways while inhibiting tumor progression, cell growth, cell proliferation, and survival.

### 4.3. Prostate Cancer (Metastasizes Mainly to Bones, Liver, Lung, and Brain)

Ahsg possesses prognostic utility as an indicator for metastatic prostate cancer, as it has been shown to aid tumor cell adhesion, spreading, and invasion. When compared to normal prostate tissue, metastatic prostate cancer shows elevated Ahsg expression. This serum immune reactivity to Ahsg was significantly higher in castrate-resistant bone-metastatic prostate cancer compared to organ-confined cancer. Increasing serum antibody reactivity to Ahsg was correlated with worse outcomes [63]. In vitro studies reveal that mechanistically, Ahsg facilitates both the anchorage-independent and anchorage-dependent growth of prostate cancer cells. Endocytic uptake of Ahsg in tumor cells leads to rapid cellular adhesion and spreading mediated by TLR4 [33]. The downstream impacts of TLR4 on cell proliferation pathways such as MAPK and P13/AKT signaling contribute to the effect of Ahsg in cancer progression. Interestingly, Ahsg has also been shown to contribute to the growth of non-adherent cells. While Ahsg is readily internalized in adhered and spread cells, it remains on the surfaces of non-adherent cells. By sequestering vesicles of various sizes on the surfaces of spheroid cells, Ahsg promotes the transmission of growth signals that stimulate the AKT and MAP kinase pathways [64]. Interestingly, 3D cell-growth signaling pathways showed prolonged activation compared to 2D growth. Thus, Ahsg demonstrates promise as a prognostic indicator for metastatic prostate cancer, as its expression correlates with tumor aggressiveness and is associated with poorer outcomes. Publicly available data (TCGA) suggest that prostate cancer cells with high ectopic synthesis of Ahsg (mRNA) have a poorer progression-free survival compared to those that express low expression [58].

### 4.4. Colorectal Cancer

Ahsg contributes to colorectal cancer pathogenesis and progression. On the one hand, Ahsg triggers the secretion of ‘adhesion competent’ exosomes containing Ahsg, plasminogen, and histones, promoting colorectal cancer (CRC) cell adhesion and spreading [29]. In mice, expression of macrophage markers and an increase in cytotoxic T lymphocyte (CTL) markers resulted in decreased numbers of large polyps, indicative of reduced intestinal tumorigenesis [58]. Since Ahsg is a macrophage chemoattractant, increased levels of Ahsg may provide a biomarker for macrophage activation, as this relates to mouse intestinal tumorigenesis; however, this link requires further research [5]. Another potential link lies in the finding that colorectal cancer pathogenesis aligns with biological aging, as colorectal cancer patients exhibit shorter telomeres compared to healthy controls. Notably, lower levels of Ahsg were associated with shorter telomeres in CRC patients, suggesting a potential link between Ahsg, redox control, calcium homeostasis, and telomere dynamics [65]. However, in contrast, a European control study found only a modest linear association between Ahsg concentrations and colorectal cancer risk, particularly in men. Mendelian randomization analysis suggested that genetically raised Ahsg may not directly contribute causally to colorectal cancer risk, indicating the need for further research to elucidate the causal relationship between Ahsg and colorectal cancer development [44]. In the same vein, Ahsg hinders TGF-β1 binding to cell surface receptors, dampens TGF-β signal transmission, and inhibits TGF-β-induced epithelial–mesenchymal transition in human colorectal cancer specimens. In 66 consecutive human colorectal cancer specimens, Ahsg levels were three times lower in tumors than in normal tissue, while the levels of other plasma proteins were unchanged. Using a mouse model of intestinal tumorigenesis, increased polyp formation and progression to adenocarcinoma were observed in mice lacking Ahsg. These mice also showed heightened TGF-β signaling and suppressed immune cell function, effects that were reversed by reintroducing Ahsg. This suggests that boosting Ahsg levels may be beneficial for patients with TGF-β-driven tumors [15].

### 4.5. Brain Cancer

Studies indicate that Ahsg plays a crucial role in the signaling pathways related to the growth, motility, and invasion of glioblastomas (GBMs) in vitro *and* in vivo, suggesting its potential as a serum biomarker for treatment. In vitro *studies of* the GBM cell line LN229 showed ectopic synthesis and secretion of Ahsg in serum-free culture to support growth. Additionally, fetuin-A knockdown in LN229 cells hindered growth, movement, invasion, and uptake of labeled exosomes, while treating LN229 cells with ASF reduced their uptake and led to senescence. The D subclone of a fetuin-A knockdown had approximately 90% less ectopic fetuin-A and experienced senescence in serum-free medium, which was mitigated in the presence of purified fetuin-A [31]. The uptake of Ahsg in GBM cells appears to be mediated by TLR4. CLI-095, a specific TLR4 inhibitor, inhibits rapid fetuin-A uptake, preventing adhesion, spread, and growth in serum-free medium, especially where there is no ectopic synthesis. Furthermore, in the presence of CLI-095, the uptake of Ahsg was significantly delayed (no uptake after 120 s), contrasting with the rapid uptake (around 60 s) observed in its absence. Incubating detached LN229 cells with CLI-095 or asialofetuin-A (ASF) markedly impaired their ability to adhere and spread on plastic, indicating that inhibiting the binding of Ahsg to TLR4 reduced the growth and spread of LN229 cells [33]. Similar results were observed in an in vivo knockdown of *Ahsg*. The proliferative invasive capacities of glioblastoma cells in vivo were significantly attenuated in mice with Ahsg knockdown compared to control mice (injected with scrambled shRNA-transfected LN229 cells) [31].

### 4.6. Osteosarcoma

Though rarer compared to other tumors such as breast and prostate cancer, Ahsg is implicated in osteosarcomas. Ahsg is an established marker of bone turnover given its involvement in bone remodeling and mineralization [66]. Via its recruitment to the plasma membrane alongside activators of mineralization such as Annexin A6, Ahsg participates in modulating mineralization processes. Interestingly, in osteosarcoma (Saos-2) cells, an increase in the displacement of cytoplasmic Ahsg to the membrane was observed compared to osteoblasts (hFOB 1.19). The distribution of Ahsg in Saos-2 cells was consistent with β-actin. In resting Saos-2 cells, both Ahsg and β-actin are uniformly distributed in the cytoplasm, whereas in stimulated Saos-2 cells, Ahsg was redistributed to the sub-membrane, where it co-localized with β-actin, a key gene in cell growth and migration [67,68]. Thus, Ahsg could potentially serve as a valuable marker in conditions related to bone mineralization, such as osteosarcoma. Furthermore, the levels of Ahsg in pediatric patients after osteosarcoma treatment were found to be similar to those of healthy controls [69]. Ahsg’s involvement in bone turnover and mineralization processes, particularly its distinctive distribution in osteosarcoma cells and its levels post-treatment, highlights its potential as a valuable marker in osteosarcoma management and monitoring. Interestingly, sarcoma cells that express high levels of the Ahsg message (RNAseq) have poor overall survival compared to those that express low levels, with a median survival of 61 months for high expression and 90 months for low expression [58].

### 4.7. Lung Cancer

Ahsg is involved in the tumor colonization of the lung by activating the PI3 kinase/Akt signaling pathway, leading to increased tumor growth and adhesion in a dose-dependent and Ca (2+)-dependent manner. The absence of Ahsg inhibited tumor growth in the lungs, highlighting its significance in the development of lung cancer and metastasis [16]. After 2 weeks, mice lacking Ahsg were free of tumors, whereas wild-type controls had metastatic nodules. It is worth noting that mice heterozygous for the Ahsg locus formed half as many colonies as homozygous WT mice. Additionally, lung colonization was reinstated by administering purified Ahsg to Ahsg-null mice [16].

Conversely, a comprehensive glycoproteomic analysis of human lung adenocarcinoma tissue in 10 patients found fetuin-A levels were frequently decreased at both the mRNA and total protein levels in cancer tissues [24]. Glycoarray analysis of Ahsg revealed the presence of mannose, galactose (Gal)/N-acetylgalactosamine (GlcNAc), β-Gal, terminal GlcNAc, and sialic acid. Mannose structures were found exclusively in Ahsg from lung cancer samples. Most if not all of the biological properties of Ahsg may be attributed to its glycan moieties, underscoring the need to clarify the roles of these Ahsg ‘glyco-codes’ in normal physiology and pathophysiology. Ultimately, more research is needed to fully understand the role of Ahsg in different lung cancers and how changes in Ahsg might factor into tumor progression. To emphasize the dichotomy in the role of Ahsg in lung cancers, in lung adenocarcinoma, high ectopic synthesis of Ahsg (RNAseq) leads to poorer overall survival compared to low ectopic synthesis (Figure 4). On the other hand, in squamous lung cancers, those that express high levels of Ahsg (RNAseq) have better overall survival, with a median survival of 61 months compared to 47 months for those that express low levels [58].

### 4.8. Head and Neck Squamous Cell Carcinomas

The first indication of an association between Ahsg and head and neck squamous cell carcinomas (HNSCCs) resulted from an HNSCC serum proteome analysis showing elevated levels of Ahsg in patients with late-stage disease. This was somewhat unexpected, in that serum Ahsg levels are quite high to begin with and decrease as a consequence of a heavy disease load (negative acute-phase response protein) [70]. Until this time, it was believed that cancer cells acquired Ahsg from the serum, but these data suggested that HNSCC cells synthesize and secrete Ahsg. The data were later corroborated in prostate cancer cells, as alluded to above [63]. Thompson et al. reported the presence of Ahsg mRNA and protein in the HNSCC cell lines SQ20B, FaDu, and UMSCC47. RNAi-mediated subline generation of the SQ20B cell line (AH50 and AH20, expressing Ahsg at 50% and 20% of wild-type levels, respectively) confirmed that Ahsg supported in vitro serum-free growth in addition to blunting other in vitro metastatic properties such as motility and invasion. Interestingly, in TCGA data, overall survival is poorer in head and neck squamous carcinoma cells that synthesize high levels of Ahsg (Figure 5) [58]. Additionally, AHSG was shown to have protected MMP-9 from autolytic degradation. In other words, in the presence of AHSG, MMP-9 remains present at levels that support the degradation of the ECM, which could result in metastasis.

### 4.9. Other Cancers

There is emerging evidence implicating Ahsg in various other cancers, including pancreatic cancer, retinoblastomas, chondrosarcoma, LY-R lymphoma, and acute myeloid leukemia, though the exact role of Ahsg remains to be conclusively determined. The early literature implicating Ahsg in several other cancers has demonstrated promising directions. When employing capillary electrophoresis–mass spectrometry to analyze the peptides in urine samples from 54 patients with pancreatic cancer (PCA) and 52 patients with chronic pancreatitis (CP), Ahsg emerged as the most significant peptide marker for PCA [50]. In contrast, a different study involving 81 pancreatic cancer (PCA) cases and 81 matched controls revealed that circulating Ahsg levels were comparable between patients and controls and were not linked to the severity of the disease [51]. Further investigations are necessary to conclusively determine whether Ahsg can be considered a prognostic biomarker in pancreatic cancer. Proteomic analysis of retinoblastomas isolated from the pooled clinical samples displayed the overexpression of Ahsg [71]. Similarly, Ahsg expression levels were significantly higher in the chondrosarcoma (CS) group compared to the chondroma group (*p* < 0.05), with higher immunopositivity rates observed in tissues with moderate or poor tumor differentiation, AJCC stage III or IV, Enneking stage II or III, and metastasis. Kaplan–Meier survival analysis revealed significantly shorter survival in patients with Ahsg immunopositivity (*p* < 0.05 or *p* < 0.001). Cox regression analysis indicated that Ahsg immunopositivity was negatively correlated with postoperative survival and positively correlated with mortality [48]. Also, in immunized mice bearing L5178Y-R (LY-R) lymphoma tumors, serum antibodies recognized an immunogenic form of Ahsg that was not detected in mice immunized with the non-tumorigenic variant LY-S or in healthy mice. It appears that the Ahsg recognized by anti-LY-R antibodies is a specific immunogenic form associated only with tumorigenic LY-R cells and may play a key role in the progression of LY-R lymphoma [72]. Finally, acute myeloid leukemia (AML) blasts from 79 consecutive patients cultured in serum-free medium revealed differences in the constitutive release of protease/protease regulators, including Ahsg, that contribute to disease heterogeneity [73]. While there is a relevant trend in which Ahsg levels are observed to be higher in various cancers, further investigations are warranted to conclusively determine Ahsg’s potential as a prognostic indicator considering its differential expression patterns and implications for disease progression observed across different cancer types.

## 5. Clinical Outlook

Given the extensive involvement of Ahsg in numerous tumors and metabolic conditions, it has been explored for utility in patient diagnosis, therapy management, and as a potential biomarker of metastatic malignancies. The use of Ahsg in the diagnosis of glioblastomas, hepatocellular cancer, lung adenocarcinoma, colorectal cancer, osteosarcoma, pancreatic cancer, and other cancers has been studied. Though Ahsg does not show any association with metastatic cancer in some of the above studies, it is reported to be a prominent peptide marker for many cancer types, aiding in the differentiation from benign conditions. In some cases, such as lung adenocarcinomas, complex glycosylation patterns were observed [24] (Figure 6).

In addition to aiding in diagnosis and prognosis, Ahsg can serve as a biomarker in identifying chemotherapy-induced side effects such as renal dysfunction [38]. Furthermore, Ahsg can serve as a useful marker in directing therapies and shows potential as a target for new therapies. The use of ASF and CLI-095 to inhibit the binding of Ahsg to TLR4 receptors significantly hindered cell adhesion and spreading in tumor cells. Furthermore, knockdown and knockout of Ahsg in mouse mammalian models further attenuated tumorigenesis and tumor progression. Ahsg, however, was shown to not be predictive of the response to chemoradiation in glioblastoma [74]. In addition, studies have also identified polymorphic variants associated with Ahsg expression in cancer or other pathologies such as diabetes [75,76].

As it is, most cancer cells in culture appear to depend on the Ahsg derived from bovine serum to drive their growth [21]. Nevertheless, some tumor cells can grow well in a serum-free medium because their growth is driven by the endogenous ectopic Ahsg that they make [31]. It is more than likely that this scenario is repeated in vivo, where some tumors in the initial stages of growth depend on the ectopic Ahsg that they make for their growth dynamics [63]. This could explain why tumor cells that express high levels of Ahsg have poorer overall survival compared to those that express low levels. On the other hand, those that depend mostly on blood-derived Ahsg may have a growth advantage when there is an adequate blood supply to the tumor and may even have a higher propensity to move into the blood vessels (intravasation), as suggested by in vitro studies [28].

## 6. Summary

In summary, Ahsg is clearly a multifunctional protein whose role in cancer biology is quickly gaining traction in the literature. The chances are that it is more relevant in the growth of cancer cells that preferentially metastasize to the bones, liver, lungs, and brain, tumor microenvironments that tend to be rich in Ahsg. Future studies should be directed towards the ectopic synthesis of Ahsg and how this might influence the progression and or prognosis of cancer cells. Ahsg’s involvement in the biogenesis and secretion of bioactive exosomes, as well as in the promotion of the 3D growth of tumor cells will also continue to open new avenues in the study of tumor cell growth and dissemination. Finally, the occurrence of SNP variants of Ahsg as well as its glycosylation status may also shed new light on its role as a tumor diagnostic or prognostic protein. These are questions that must be addressed in the future.

## Figures and Tables

**Figure 1 ijms-25-12918-f001:**
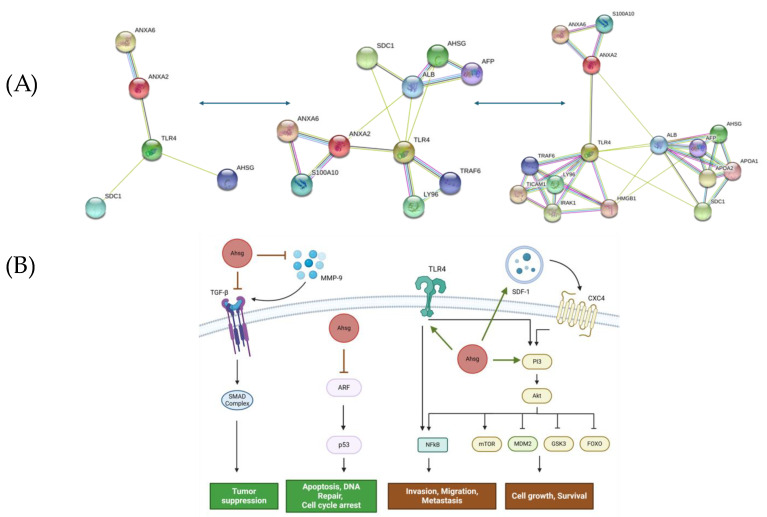
Ahsg-interacting proteins in cancer pathology. (**A**) The interactive network of Ahsg and its target proteins highlights the protein–protein interactions with known and predicted binding partners in humans. The network includes key interactions with Annexin-A2 (ANXA2), Toll-like receptor 4 (TLR4), and Syndecan-4 (SDC4), among others. The network provides insight into Ahsg’s involvement in signaling pathways related to tumor growth and survival. (**B**) Ahsg participates in diverse physiological processes and is linked to tumor growth and spread. It interacts with multiple proteins, impacting signaling pathways crucial for cancer advancement, including tumor cell proliferation, migration, and invasion. In particular, Ahsg activates (green arrows) pro-tumor pathways and inhibits (red inhibition arrows) anti-tumor pathways. These interactions contribute to both the pro-tumor and anti-tumor effects of Ahsg in cancer.

**Figure 2 ijms-25-12918-f002:**
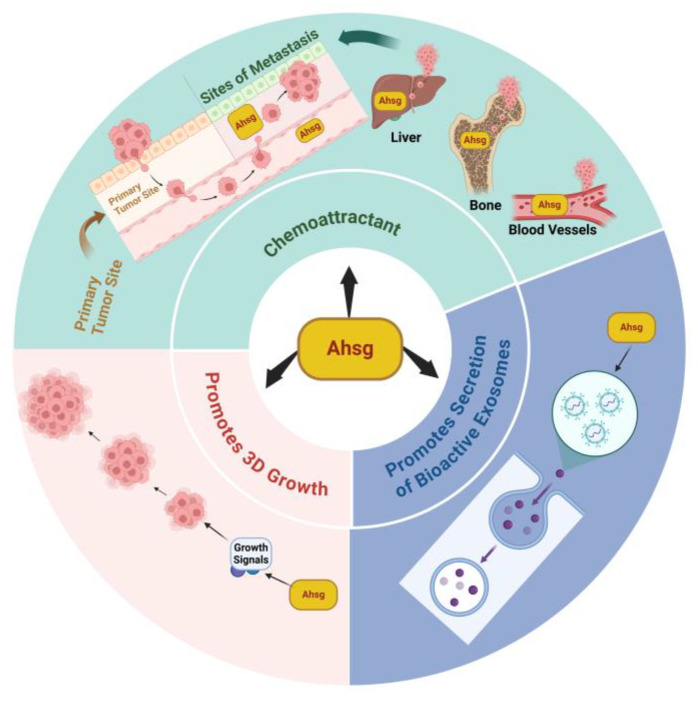
Ahsg promotes key processes in cancer cell growth and dissemination: Ahsg promotes the secretion and update of bioactive exosomes, 3D growth, and the homing of tumor cells to bone and liver.

**Figure 3 ijms-25-12918-f003:**
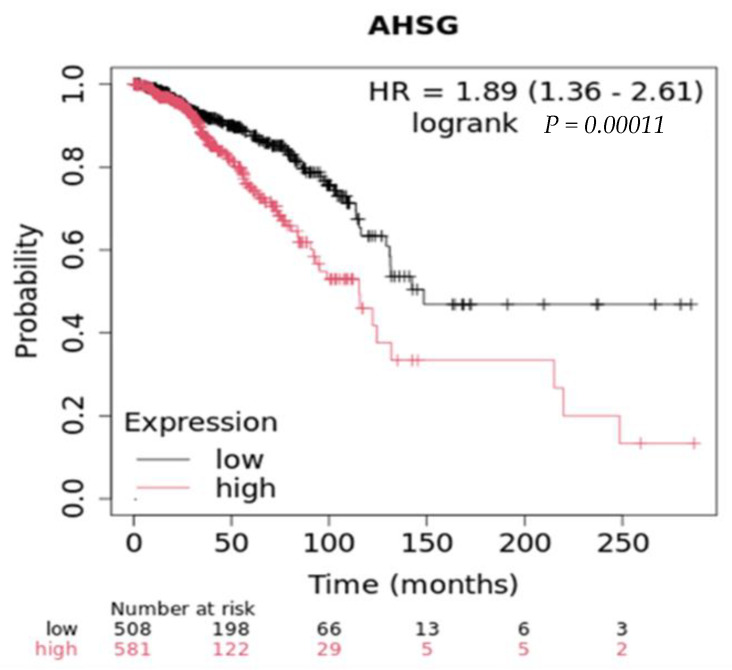
Overall survival in in breast cancers with low and high expression (RNAseq) of Ahsg.

**Figure 4 ijms-25-12918-f004:**
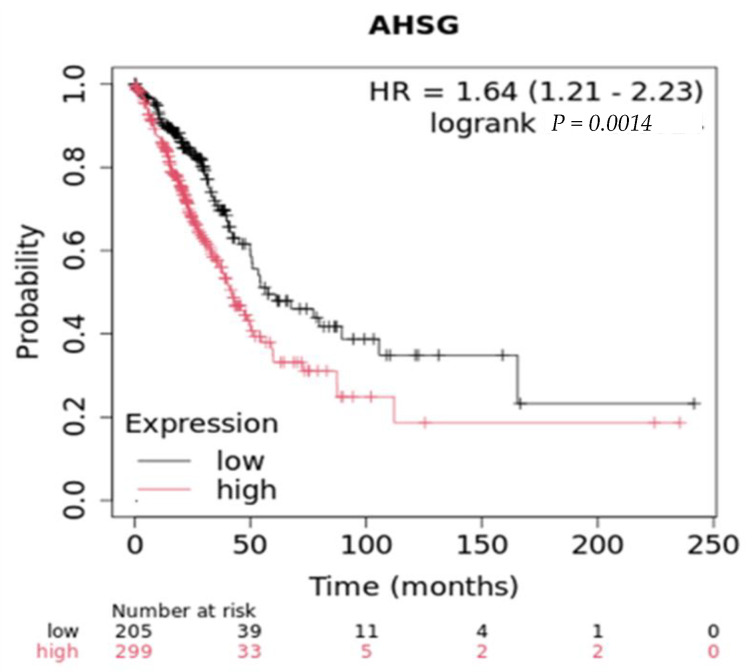
Overall survival in lung adenocarcinomas with low and high expression (RNAseq) of Ahsg.

**Figure 5 ijms-25-12918-f005:**
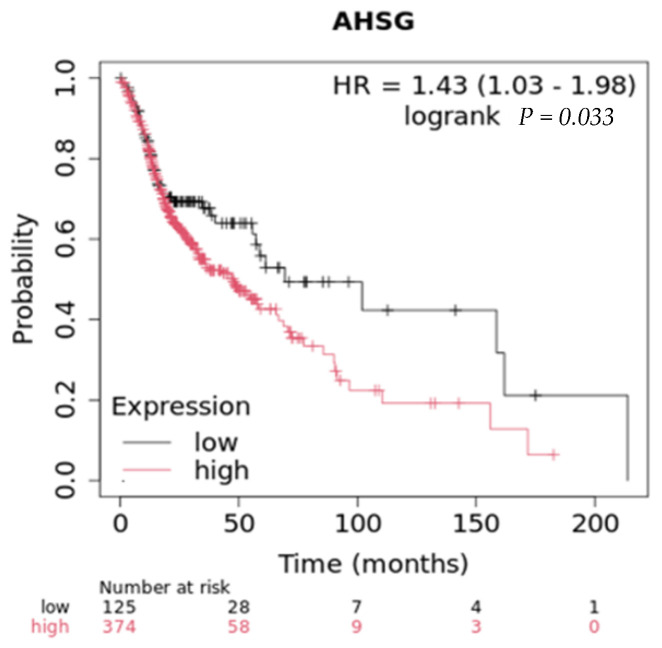
Overall survival in head and neck squamous carcinomas with low and high Ahsg expression (RNAseq).

**Figure 6 ijms-25-12918-f006:**
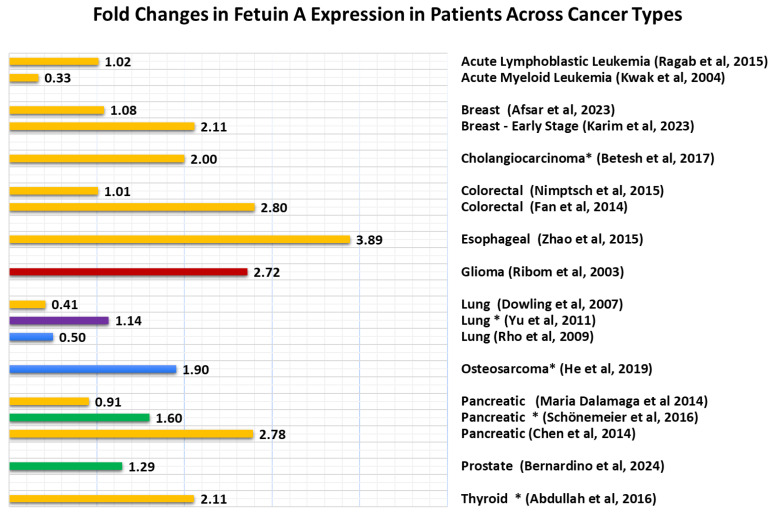
Fold changes in Ahsg expression in patients across cancer types. Ahsg expression levels in cancer patients are variable across the literature. A comparison of the fold increases and decreases (cancer/normal) in Ahsg in cancer patients relative to controls reveals that Ahsg levels are upregulated in most cancer patient cohorts. Studies that revealed a downregulation in Ahsg in cancer patients are also noted with fold changes <1. The studies compared included those on leukemia [37,38], breast cancer [39,40], cholangiocarcinoma [41], colorectal cancer [42,43], esophageal cancer [44], glioma [17], lung cancer [24,45,46], osteosarcoma [47], pancreatic cancer [48,49,50], prostate cancer [51] and thyroid cancer [52]. The variability in fold change is demonstrated both across cancer types and between studies of patients with the same cancer type. * Studies compared cancer patients to benign-disease controls rather than healthy controls. References are provided.

## Data Availability

The data supporting Figure 1A, which illustrate protein interactions with Ahsg, are openly accessible through the STRING database at https://version-12-0.string-db.org/cgi/network?networkId=b1dmamQNY4WG (accessed on 20 November 2024). Additionally, the data presented throughout this manuscript were obtained from an extensive literature review, with all referenced sources properly cited within the text and indicated in square brackets within the figures to attribute them to the original studies.

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
