# Peer review of "The Role of Fetuin-A in Tumor Cell Growth, Prognosis, and Dissemination"

_ijms, 2024, doi:10.3390/ijms252312918_

Round 1

Reviewer 1 Report

Comments and Suggestions for Authors

The manuscript presented by Odiase et al. offers a comprehensive review of the role of Fetuin-A across various cancer types, highlighting its involvement in signaling pathways, invasion, metastasis, and its potential as a diagnostic and therapeutic biomarker. However, I believe the review could benefit from additional contextualization in the introduction to help readers understand the clinical relevance of Fetuin-A in comparison to other biomarkers.

Major Comments

-A comparison of Fetuin-A with other oncological biomarkers would clarify whether its specificity and sensitivity are superior or complementary.

-The review extensively covers in vitro and animal model studies but lacks clinical studies in humans that illustrate practical applications of Fetuin-A. A dedicated section on current or developing clinical studies would be beneficial.

-A more detailed discussion is needed on the challenges of implementing Fetuin-A in clinical settings. This could address variability in Fetuin-A levels according to cancer stage or non-cancer-related factors.

-The title is too general and should be restructured to reflect the manuscript's content better. Suggested titles could include “Roles in Tumor Growth, Metastasis, and Clinical Potential” or “Molecular Mechanisms to Diagnostic and Therapeutic Applications.”

-The review includes four figures; Fig. 1 is appropriate, but additional figures illustrating tumor suppression, invasion, metastasis, etc., would be valuable (see biorender pre-images). A comparative figure showing Fetuin-A expression levels in different cancer types and its relevance in tumor survival and progression would highlight the contextual effects of Fetuin-A and complement data in Figs. 2–4.

-An interactive network of Fetuin-A with its target proteins (such as annexin-A2, TLR4, and SDC4) would improve comprehension of these interactions.

I would suggest including a study to strengthen the context of Fetuin-A in lung cancer:

Glycoproteomic Analysis in Lung Cancer: DOI: 10.1007/s10930-009-9177-0

Overall, the manuscript provides an in-depth analysis of Fetuin-A in oncology but would benefit from an expanded discussion on the clinical implications of its use and a comparison with other biomarkers.

Author Response

-A comparison of Fetuin-A with other oncological biomarkers would clarify whether its specificity and sensitivity are superior or complementary.

Reviewer 2 Report

Comments and Suggestions for Authors

- The paper would be strengthened with a more targeted conclusion highlighting the most significant and consistent findings and knowledge gaps.

- The review focused on some cancers and ignored other types such as head and neck, other cancers.

- More visual aids, like schematic diagrams, tables, or figures, could improve the review by better illuminating the main mechanisms and conclusions covered.

- Although fetuin-A promotes cancer growth, fetuin-A has a pro-apoptotic effect. This part should be discussed and the mechanism behind this uncovered.

Author Response

Comment 1.  The paper would be strengthened with a more targeted conclusion highlighting the most significant and consistent findings and knowledge gaps.

Response: We thank the reviewer for this suggestion and have now included a summary of the most significant and consistent findings and knowledge gaps (See page 11).

Comment 2- The review focused on some cancers and ignored other types such as head and neck, other cancers.

Response:  We thank the reviewer for this suggestion.  We have now included a section on head and neck and other cancers (Pages 8 and 9)

Comment 3- More visual aids, like schematic diagrams, tables, or figures, could improve the review by better illuminating the main mechanisms and conclusions covered.

Response: We thank the reviewer for this suggestion. We have now included 2 more figures (Fig. 2 and Fig. 6)

Comment 4- Although fetuin-A promotes cancer growth, fetuin-A has a pro-apoptotic effect. This part should be discussed and the mechanism behind this uncovered.

Response: We thank the reviewer for this suggestion.  We have now discussed this on page 2, green letters.

Reviewer 3 Report

Comments and Suggestions for Authors

The article "The Role of Fetuin-A in Cancer" reviews the role of Fetuin-A (Ahsg), a glycoprotein, in liver, breast, prostate, lung and other cancers. Fetuin-A has multiple functional roles in cancers: it promotes cell proliferation, metastasis, and invasion of other tissues through intracellular signaling pathways, cytokine regulation, and inflammation induction. It is related to the binding and adhesion properties of cancer cells, enhancing their potential for invasion and migration. Fetuin-A is proposed as a potential biomarker for the diagnosis and prognosis of various types of cancer. In my opinion, the article is important as it presents Fetuin-A as a multidimensional molecule that is actively involved in many neoplastic processes and is therefore promising for further research and potential clinical use, as well as a diagnostic biomarker for a variety of cancers.

The article could be published if some important issues are addressed:

-Separation by cancer type: Although the article examines many different cancer types, the structure could be improved by categorizing the cancer types into distinct sections to clarify the contribution of Fetuin-A (Ahsg) in each cancer type.

-Summary of Conclusions: A summary section can be added at the end of the article summarizing the ways in which Fetuin-A affects cancer progression — reducing inflammation, increasing cell proliferation, regulating TGF-β, etc. This will provide a unified overview on the role of Fetuin-A and will make its importance clear.

-Directions for Future Research: Directions suggesting specific questions or hypotheses for future research, such as the clinical implications of variable Fetuin-A glycosylation in cancer cells or its role in different stages of carcinogenesis, could be added.

-Simplification of Technical Terms: Some terms, such as “PI3-kinase/Akt pathway” or “TGF-β signaling,” may not be readily understood by all readers. Perhaps a comprehensible explanation in parentheses, a footnote or a box would be helpful.

-Avoiding Repetition of Information: Repetitions on the mechanisms of Fetuin-A can be consolidated, such as the frequent reference to the PI3-kinase/Akt pathway at various points in the article. If the report is condensed into one section, which presents the basic mechanisms of Fetuin-A, unnecessary repetitions will be reduced, making the article easier to read.

-Comments on the Use of Fetuin-A as a Marker: The use of Fetuin-A as a marker for diagnosis or treatment in specific cancers could be further developed in modern clinical practice. A framework describing how Fetuin-A could be useful in clinical scenarios, such as in protocols to control inflammation or metastasis, would add practical value to the study.

Author Response

Comment -Separation by cancer typeAlthough the article examines many different cancer types, the structure could be improved by categorizing the cancer types into distinct sections to clarify the contribution of Fetuin-A (Ahsg) in each cancer type.

Response:  We thank the reviewer for this comment.  We have now categorized the cancer types into distinct sections to clarify the contribution of Ahsg.  We have pointed those tumor types that preferentially metastasize to the bones, liver, lungs and brain representing tumor microenvironments that tend to be rich in Ahsg.

Comment-Summary of Conclusions: A summary section can be added at the end of the article summarizing the ways in which Fetuin-A affects cancer progression — reducing inflammation, increasing cell proliferation, regulating TGF-β, etc. This will provide a unified overview on the role of Fetuin-A and will make its importance clear.

Response:  We thank the review for this comment that was also pointed by another reviewer.  We have now added a summary section at the end of the manuscript-blue letters.

Comment-Simplification of Technical TermsSome terms, such as “PI3-kinase/Akt pathway” or “TGF-β signaling,” may not be readily understood by all readers. Perhaps a comprehensible explanation in parentheses, a footnote or a box would be helpful.

Response: We thank the reviewer for this comment.  We have now explained these terms (see page 2, green letters).

Comment-Avoiding Repetition of InformationRepetitions on the mechanisms of Fetuin-A can be consolidated, such as the frequent reference to the PI3-kinase/Akt pathway at various points in the article. If the report is condensed into one section, which presents the basic mechanisms of Fetuin-A, unnecessary repetitions will be reduced, making the article easier to read.

Response:  We thank the reviewer for this suggestion and have removed some of frequent references to PI3-kinase/AKT.

Round 2

Reviewer 1 Report

Comments and Suggestions for Authors

I consider that the authors have improved their revision, and the new figures effectively complement the manuscript's message. However, I still have a few minor observations:

-Standardize the terminology for Fetuin-A or Ahsg throughout the manuscript. Decide whether to use FetA, Ahsg, or FetA/Ahsg consistently.

-Accordingly, update Fig. 1 to reflect this change and ensure it reads FetA, Ahsg, or FetA/Ahsg.

-For Fig. 1 and Fig. 6, the Data Availability Statement must include the respective URLs for the String public database and the database of fold changes in Fetuin-A/Ahsg expression across cancer types.

Author Response

-Standardize the terminology for Fetuin-A or Ahsg throughout the manuscript. Decide whether to use FetA, Ahsg, or FetA/Ahsg consistently.

Response:  We thank the reviewer for this comment.  We have now standardized our terminology and are now using Ahsg throughout the manuscript.

-Accordingly, update Fig. 1 to reflect this change and ensure it reads FetA, Ahsg, or FetA/Ahsg.

Response: We have updated Fig. 1 as suggested.

For Fig. 1 and Fig. 6, the Data Availability Statement must include the respective URLs for the String public database and the database of fold changes in Fetuin-A/Ahsg expression across cancer types.

Response: We have included the URL for String public database.

Reviewer 2 Report

Comments and Suggestions for Authors

The authors followed the comments.

Comments on the Quality of English Language

English is fine.

Author Response

 We thank the reviewer for letting us know that we adequately addressed their concerns.

Reviewer 3 Report

Comments and Suggestions for Authors

The authors made all corrections. The revised version is completely agreeable to me, for this reason, I recommend the publication of the manuscript.

Author Response

(The authors gave the same response as above.)
